# A Yin–Yang Framework for Understanding Regional Cultural Dynamics: Insights from the Three Kingdoms of China

## Abstract

This paper develops a theoretical-conceptual framework for analyzing intra-national cultural variation in China by applying the Yin–Yang model as an anti-essentialist lens. Whereas most existing cultural geography and cross-cultural studies rely on static typologies, the Yin–Yang approach foregrounds dynamic balance, historical contingency, and the co-presence of complementary tendencies. The Three Kingdoms macro-regions (Wei, Shu, Wu) are employed not as historical case studies per se, but as a *heuristic illustration* to demonstrate how regional cultural identities can be theorized as oscillating configurations of openness and consolidation, continuity and transformation. By integrating insights from philosophy, intercultural studies, and historical geography, the paper extends Yin–Yang theory beyond cross-national applications and positions it as a methodological alternative to dichotomous frameworks such as Hofstede's cultural dimensions. The contribution lies not in empirical testing but in conceptual advancement: showing how Yin–Yang can structure regional cultural analysis while safeguarding against stereotyping. The framework has implications for cultural theory, comparative geography, and intercultural methodology more broadly.

## 1 Introduction

China's vast cultural landscape resists reduction to uniformity. Across millennia, communities have adapted to sharply divergent geographies, ecological conditions, and political regimes. Such diversity is not merely spatial; it unfolds historically through shifting balances of power, trade, and cultural synthesis. Conventional approaches to regional culture—whether in intercultural communication, anthropology, historical geography, or area studies—have tended toward static typologies. These often risk stereotyping, by presenting cultural differences as immutable "traits" rather than historically contingent patterns (Fang, 2012). This paper advances an alternative approach rooted in the Yin–Yang conceptual framework, applying it to three historically resonant macro-regions derived from the Three Kingdoms period: Wei (North-Central Plain), Shu (Sichuan Basin), and Wu (Lower Yangtze/Yangtze Delta). The Yin–Yang lens conceptualizes cultural variation as a dynamic interplay of complementary and opposing forces—stability and change, centripetal and centrifugal tendencies, consolidation and innovation—rather than fixed polarities (Fang, 2012). This dynamic model accommodates both continuity and transformation, offering a means to describe complexity without resorting to essentialism.

### 1.1 Research gap and research question

While Yin–Yang has been increasingly adopted in cross-cultural management and organizational studies (Fang, 2012), its systematic application to intra-national regional cultural variation—particularly

Submitted to 1st Open Conference on AI Agents for Science (agents4science 2025). Do not distribute.

in China—remains underdeveloped. Existing cultural geography tends to classify regions via economic indicators, linguistic groupings, or historical political boundaries which is dominating (Chiang, 2005), often without a framework for integrating complementary contradictions. This disconnect between more complex conceptual framing and empirical regional studies is the central gap this paper addresses. Thus, the present study asks: *How can the Yin–Yang framework be applied to analyze and interpret regional cultural variation in China, using the Three Kingdoms macro-regions as a heuristic, in ways that avoid stereotyping and integrate historical, linguistic, and material evidence?* This paper contributes to theory by extending the Yin–Yang framework from its common use in cross-national management studies to the analysis of intra-national regional cultural dynamics. The contribution is twofold: first, it develops Yin–Yang as a methodological safeguard against essentialist cultural typologies by conceptualizing regional traits as historically contingent balances rather than fixed attributes. Second, it demonstrates how a heuristic case (the Three Kingdoms macro-regions) can operationalize this lens in cultural geography, offering a transferable model for studying other internally differentiated world regions.

## 1.2 Societal relevance

Understanding China's regional cultural complexity has tangible implications beyond academia. In domestic policy, nuanced regional analyses can inform balanced development strategies that account for local socio-cultural strengths and needs. In education, teaching about diversity within China—rather than portraying a monolithic culture—fosters intercultural competence, both domestically and internationally (Goh, 2012). For diplomacy and global business, this approach helps prevent miscommunication rooted in overgeneralization, offering a richer basis for partnership building. Finally, heritage preservation initiatives benefit from recognizing the dynamic hybridity of regional traditions, ensuring that both continuity and adaptive innovation are valued.

# 2 Theoretical framework: Yin–Yang as an analytical lens

## 2.1 Philosophical origins and classical interpretations

The concept of Yin-Yang (陰陽/阴阳) occupies a central position in classical Chinese thought, appearing in early cosmological and medical texts such as the *Yijing* (易經/易经, *Book of Changes*) and the *Huangdi Neijing* (黃帝内經/黄帝内经, *Yellow Emperor's Inner Canon*). Yin is conventionally associated with receptivity, consolidation, and inward movement, while Yang is linked to activity, expansion, and outward movement (Li, 2014). Crucially, the two are not opposites in the Western dialectical sense but mutually constitutive: each contains the seed of the other, as represented by the small contrasting dot in the *taiji* (太極/太极) diagram. Han dynasty cosmology emphasized the cyclic transformation between Yin and Yang as a model for natural and social change. This philosophical basis allows for the conceptualization of culture not as a static "essence," but as an evolving balance of complementary tendencies.

## 2.2 Contemporary adoption in cultural studies

In recent decades, Yin–Yang has been mobilized as a meta-theoretical framework in cross-cultural management, organizational theory, and intercultural communication (Fang, 2012; Li, 2014). Fang (2012) proposes the Yin–Yang perspective as a counterpoint to Hofstedean cultural dimensions, which risk oversimplifying by fixing cultures along static bipolar scales. Instead, Yin–Yang acknowledges that "opposite" traits may co-exist and transform into one another over time. Applied to intra-national variation, this model permits a dynamic mapping of regional differences. Rather than categorizing a region as permanently "collectivist" or "individualist," for instance, the Yin–Yang approach can identify periods and contexts in which one tendency is more pronounced, and how that shifts in interaction with others.

## 2.3 Methodological implications for regional cultural analysis

Integrating Yin–Yang into regional studies involves three key methodological commitments: 1. Relational definition: Traits are defined by their position relative to others, not in isolation (Fang, 2012); 2. Balance recognition: Both stabilizing (Yin) and transformative (Yang) forces are present in each

case, though their proportions vary (Li, 2014); 3. Historical contingency: The balance is temporally situated; shifts over decades or centuries must be traced (Fang, 2012). In practical terms, this requires a multi-source approach that combines: 1. Historical analysis of political, economic, and social structures (De Crespigny, 2019; Farmer, 2019; Zhi, 1998); 2. Archaeological and material culture evidence, which often preserves regional specificities beyond textual sources (Kyushu National Museum, 2019); 3. Linguistic evidence to trace cultural exchange and divergence (Huang et al., 2024). This integrated design allows for a new heuristic theory application underlined with diverse empirical anchors.

## 2.4 Yin–Yang as anti-essentialist safeguard

One of the risks in regional cultural studies is falling into stereotype traps—treating regional patterns as immutable "character traits" of peoples. The Yin–Yang model, by embedding the expectation of change within its core logic, acts as an epistemic safeguard. A region's "inwardness" (Yin) in one historical period may transform into "openness" (Yang) under different geopolitical or ecological conditions. Conversely, phases of outward dynamism may be followed by consolidation and inward focus. In this way, Yin–Yang serves both as an analytical framework and as a methodological ethic: to remain attuned to contradiction, transformation, and the co-presence of apparent opposites in any given cultural setting.

## 3 Historical-geographical basis: The Three Kingdoms as heuristic

### 3.1 The Three Kingdoms as a cultural macro-map

The Three Kingdoms period (220–280 CE) is among the most storied eras in Chinese historiography, memorialized in Chen Shou's *Sanguozhi* (三國志/三国志) and later romanticized in *Romance of the Three Kingdoms*. Although politically transient, the tripartite division into Wei, Shu, and Wu corresponds to enduring macro-geographical configurations: 1. Wei—centered in the North China Plain and Loess Plateau, controlling key grain-producing basins and major river corridors (The Editors of Encyclopaedia Britannica, 2023c); 2. Shu—the Sichuan Basin, bounded by high mountains, controlling upper Yangtze routes and sheltered agricultural heartlands (The Editors of Encyclopaedia Britannica, 2023b); 3. Wu—the Lower Yangtze and coastal regions, connected to maritime trade networks and fertile delta plains (The Editors of Encyclopaedia Britannica, 2023a). While subsequent dynasties repeatedly reconfigured administrative borders, these three zones persisted as recognizable cultural-geographical units, their identities reinforced by topography, hydrology, and infrastructure patterns which will be shown in the next steps.

### 3.2 Environmental determinants of cultural divergence

Wei's open northern plains facilitated large-scale cereal agriculture and the rapid mobilization of cavalry-based armies. In Yin-Yang terms, this openness (Yang) in military and trade logistics was balanced by a tendency toward bureaucratic centralization (Yin) to manage vast territories (De Crespigny, 2019). Shu, enclosed by mountain ranges and accessed mainly via narrow passes such as Jianmen (劍門關/剑门关), developed a cultural orientation toward internal stability and resource self-sufficiency (Yin), periodically counterbalanced by strategic bursts of outward engagement through controlled riverine corridors (Yang) (Farmer, 2019). Wu, straddling riverine and maritime zones, exhibited a long-standing duality: cosmopolitan openness to maritime exchange (Yang) and the cultivation of refined artistic and literary traditions that reinforced a cohesive regional identity (Yin) (Zhi, 1998).

### 3.3 Infrastructure and communication patterns

Local recommendation models (Zhao et al., 2023) reveal that ancient transportation systems both reinforced and transcended these macro-regions. For instance, Shu's internal waterway network promoted intra-basin cohesion, while Wu's port complexes connected the lower Yangtze to East Asian and Southeast Asian circuits. Wei's strategic control of the north–south trunk routes allowed for rapid administrative integration but also exposed it to repeated nomadic incursions—forcing periodic defensive consolidation. These infrastructural patterns shaped information flow, economic

integration, and cultural diffusion in ways that mirror the Yin–Yang alternations described earlier: periods of expansive interaction alternated with phases of defensive retrenchment, often in response to ecological shocks or political instability (De Crespigny, 2019).

### 3.4 The heuristic value of the Three Kingdoms division

While the Three Kingdoms period lasted less than a century, its tripartite structure serves as a heuristic macro-map for studying regional cultural variation. It offers: 1. Geographical coherence—each zone aligns with distinct ecological and infrastructural systems; 2. Historical persistence—regional differentiation can be traced through multiple dynasties, even as political borders shifted; 3. Cultural resonance—the Three Kingdoms narrative remains deeply embedded in Chinese popular conscious-ness, providing a shared cultural reference point that continues to inform identity discourses (Besio & Tung, 2008). From a Yin–Yang perspective, these macro-regions can be analyzed as historical fields of dynamic balance—not fixed "characters," but shifting equilibria of openness and closure, centralization and decentralization, interaction and isolation.

## 4 Regional profiles in the Yin–Yang Model

### 4.1 Wei: Administrative gravitation versus mobilizing openness

The Wei domain, anchored in the North China Plain and extending into the Loess Plateau, occupied the geopolitical heartland of premodern China. The vast flatlands facilitated large-scale grain cultivation, particularly of wheat and millet, and supported the rapid deployment of cavalry and chariot forces (De Crespigny, 2019). These features correspond to Yang tendencies—outward projection of military power, logistical mobility, and territorial expansion. However, sustaining control over this broad and exposed territory required a counterbalancing Yin force: the centralization of administrative authority. Wei's bureaucratic institutions, inherited and adapted from the late Han dynasty, standardized taxation, codified law, and established an intricate hierarchy of prefectures and counties (De Crespigny, 2019). This administrative gravitation created internal cohesion but also a certain rigidity in governance. From a Yin–Yang lens, Wei exemplifies the oscillation between openness and consolidation. Periods of rapid military expansion were often followed by retrenchment phases, during which infrastructural maintenance and bureaucratic oversight dominated. This balance is evident in both historical chronicles (Chen Shou, *Sanguozhi*, 三國志) and in archaeological patterns of settlement expansion and contraction (Zhao et al., 2023).

### 4.2 Shu: Enclosed continuity versus creative intensity

The Sichuan Basin, Shu's territorial base, presents one of the most geographically enclosed regions in China. Surrounded by high mountain chains such as the Qinling and Daba ranges, Shu's access to external regions was largely funneled through narrow passes and river gorges. This Yin orientation fostered a high degree of self-sufficiency: a stable agricultural economy centered on rice and diverse horticulture, coupled with a robust local craft industry (Farmer, 2019). Yet, enclosure also acted as an incubator for Yang bursts of creativity. Isolated from some external pressures, Shu developed distinctive cultural forms, from localized bronze traditions (Farmer, 2019) to innovative military engineering, exemplified by Zhuge Liang's (諸葛亮/诸葛亮) logistical inventions. When strategic openings arose—for instance, through temporary control of the upper Yangtze—Shu projected considerable cultural and political influence outward, albeit in concentrated, time-bound episodes. The Yin–Yang rhythm here is less about alternating large-scale expansion and retrenchment, and more about punctuated creativity emerging from sustained continuity. Linguistic phylogenetic studies suggest that southwestern Mandarin dialects, influenced by indigenous languages, retain features that diverge significantly from northern varieties (Huang et al., 2024), a testament to the region's capacity for cultural hybridization.

### 4.3 Wu: Cultivated refinement versus maritime openness

The Wu polity occupied the lower Yangtze basin and adjacent coastal regions, an ecological zone characterized by fertile alluvial soils and dense river networks leading to the East China Sea (Zhi, 1998). This geography facilitated intensive wet-rice agriculture and supported urban centers engaged

in long-distance trade. Wu's Yang aspect manifested in its maritime orientation: evidence shows robust participation in regional exchange networks extending to the Korean Peninsula, Japan, and Southeast Asia (Zhi, 1998). At the same time, Wu cultivated a distinctive Yin identity rooted in literati traditions, refined artistic production, and elite patronage of poetry, calligraphy, and music. This cultural refinement served as both a marker of status and a cohesive force binding regional elites (Zhi, 1998). In the Yin–Yang dynamic, Wu appears as a zone where external openness and internal cultivation coexisted symbiotically, each reinforcing the other: maritime wealth financed artistic production, while the prestige of refined culture enhanced Wu's diplomatic standing.

## 4.4 Comparative synthesis

Using Yin–Yang as a comparative lens reveals that all three regions balanced openness and consolidation, but the forms of these balances differed: 1. Wei: large-scale oscillations between military expansion and bureaucratic retrenchment; 2. Shu: sustained continuity punctuated by concentrated bursts of innovation; 3. Wu: simultaneous cultivation of internal refinement and external maritime engagement. These distinctions matter because they resist essentialist "character" labels (e.g., "northern aggressiveness" or "southern sophistication") by showing how each region contains both Yin and Yang elements in unique configurations, shifting across historical periods.

# 5 Interdisciplinary evidence

## 5.1 Dialectometry and regional differentiation

The diversification of Chinese dialects offers a linguistic mirror of the historical-geographical divisions outlined in sections 3 and 4. Located in the field of dialectometry, Huang et al. apply computational methods to trace the evolution of geo-linguistic dialect classification, and reveal distinct clusters corresponding to northern and southern speech zones (2024). These patterns somewhat align with the territorial footprints of Wei, Shu, and Wu, suggesting that linguistic divergence has deep historical roots shaped by both environmental boundaries and migration flows. For heuristic purposes, this three-fold distinction is applied here which overlaps with Huang et al.'s (2024) early classification stages. While they differentiated ten dialect systems in total, on a more general level three were distinguished: In the Wei heartland, Northern Mandarin dialects display relatively low internal diversity, reflecting a history of sustained political unification and high mobility across the North China Plain. In Shu, Southern Mandarin incorporates features from Tibeto-Burman languages, evidence of long-term contact with non-Han populations within a geographically enclosed basin. Southeast dialects—corresponding with Wu—meanwhile, show high tonal complexity and distinctive phonotactics, likely reinforced by maritime trade contacts and urban cosmopolitanism in the lower Yangtze. Seen through a Yin–Yang lens, the relative homogeneity of Wei and Shu stems from different balances: in Wei, administratively driven consolidation (Yin) is periodically energized by mobility and integration (Yang) without eroding a stable linguistic core; in Shu, geographic enclosure sustains Yin continuity, and episodic Yang openings along river corridors likewise leave internal coherence largely intact. By contrast, Wu's maritime crossroads amplifies Yang openness and multidirectional exchange, producing pronounced dialectal diversity that localizing Yin practices only partially temper. Thus, similar outcomes in Wei and Shu (homogeneity) arise from distinct Yin–Yang configurations, whereas Wu exemplifies a diversity-producing Yang tilt.

## 5.2 Evidence from archaeology

The Kyushu National Museum's special exhibition *The Three Kingdoms* (October 1, 2019–January 5, 2020) provided an in-depth exploration of the Wei, Shu, and Wu states, which contended for supremacy in China between 220 and 280 CE. The exhibition illuminated the distinctive political, cultural, and military characteristics of each kingdom. Artifacts from Cao Cao's mausoleum exemplified the centralized authority and martial rigor of the Wei state, reflecting the consolidation of power under its preeminent leader. In contrast, Shu's legacy of loyalty, righteousness, and heroic idealism was embodied in life-sized statues of figures such as Guan Yu and Zhang Fei, which underscored the moral and literary significance attributed to this kingdom. The Wu state, situated in the south, manifested its emphasis on maritime prowess and courtly sophistication through treasures from the Shangfang royal mausoleum, revealing the cultural refinement and regional distinctiveness of Wu's ruling elite. Collectively, these artifacts offered not only a vivid portrayal of the inter-kingdom conflicts but also a

nuanced appreciation of the divergent sociopolitical values and aesthetic sensibilities that defined the Three Kingdoms period (Kyushu National Museum, 2019). From a Yin–Yang perspective, these flows suggest that material culture is not a static "regional tradition" but a dynamic field where openness (Yang) and retention (Yin) interact. For instance, Wu's artifacts emerge from both external inputs and selective local adaptation, while Shu's relics are punctuated by bursts of stylistic innovation coinciding with periods of political expansion.

## 5.3 Transportation routes, intangible cultural heritage, and information flow

During the era of the Three Kingdoms (220–280 CE), the ancient Qin-Shu roads served as critical conduits for cultural exchange across the fragmented states of Wei, Shu, and Wu. Liu et al. (2022) show that intangible cultural heritage (ICH) along these routes—including folk music, traditional crafts, and rituals—was deeply influenced by the political and military dynamics of the period as well as environmental cycles. In particular, Shu, located in the Sichuan Basin, became a local center for craft and ceremonial traditions, while the movements of people and armies along the roads facilitated the diffusion of customs from Shu to neighboring regions. The study highlights how the Three Kingdoms' political divisions created both hubs and gaps in cultural transmission, leaving a lasting imprint on the spatial distribution of intangible cultural heritage that can still be traced along the Qin-Shu corridor today. In Yin–Yang terms, there is a temporal evolution beyond the Three Kingdoms period. The development of intangible cultural heritage along the Qin-Shu roads follows a "three rising and three falling" pattern: flourishing periods during the Qin and Han Dynasties, Sui, Tang, and Five Dynasties, and Ming and Qing Dynasties and slower development during the Wei, Jin, Southern and Northern Dynasties, Song and Yuan Dynasties, and modern times.

# 6 Avoiding stereotypical attributions

## 6.1 The pitfalls of static typologies

Scholarship on Chinese regional cultures has historically risked producing overgeneralized cultural "types": for example, depicting northern Chinese as inherently martial and pragmatic, or southern Chinese as inherently refined and commercially minded. While such typologies may emerge from genuine historical patterns, they become problematic when reified into essentialist identities detached from historical contingency and internal variation. This danger is amplified when regional descriptions rely heavily on selective historical episodes rather than longitudinal, interdisciplinary evidence. Thus, profound anti-essentialist, empirically informed concepts and studies are needed (for earlier approaches in this field see also the works of Fei, 1992; Zhi, 1998). The Yin–Yang framework inherently resists static categorization because it conceptualizes opposites as interdependent and in flux. A region that is militarily expansive (Yang) in one era may adopt a defensive and consolidatory posture (Yin) in another, depending on political, environmental, and technological conditions.

## 6.2 Yin–Yang as an anti-essentialist tool

Unlike dichotomous models in Western cultural analysis—e.g., Hofstede's "individualism vs. collectivism"—Yin–Yang is non-binary in function: each pole contains the seed of its opposite, implying that traits are context-dependent, co-existing, and cyclical (Fang, 2012; Peng & Nisbett, 1999). This flexibility is particularly suited to modeling regional China because: 1. Historical dynamism: Regions like Wei, Shu, and Wu display shifting balances over decades, responding to wars, trade opportunities, and climate change; 2. Internal diversity: Within each macro-region, subregions and urban centers may diverge dramatically from the regional "average"; 3. External interconnectedness: Flows of people, goods, and ideas mean that no region is culturally sealed, even in times of political isolation. In this sense, Yin–Yang does not freeze a region's identity but captures its oscillating states, making it harder for analysts to lapse into reductive cultural shorthand.

## 6.3 Methodological safeguards against stereotype formation and implications for comparative cultural research

To operationalize Yin–Yang effectively in future regional studies, several methodological safeguards are recommended: 1. Longitudinal datasets: Employ multi-decade or -century timelines to track

shifts in Yin–Yang balances, rather than single-period snapshots; 2. Multi-scalar analysis: Move between macro-regional, subregional, and local levels to capture internal diversity; 3. Interdisciplinary cross-checking: Triangulate research between linguistic, archaeological, and environmental evidence to prevent overreliance on any one dataset; 4. Discourse sensitivity: Critically assess how historical and modern narratives themselves embed stereotypes. These safeguards ensure that Yin–Yang is not just a metaphor but a structured analytical framework that disciplines interpretation.

Using Yin–Yang to analyze regional diversity in China can inform comparative work beyond East Asia. Many world regions—from the Mediterranean to the Indian subcontinent—exhibit internally differentiated yet interconnected subregions whose identities oscillate over time. In this sense, Yin–Yang offers a portable epistemological tool for resisting stereotypes in global cultural geography (Li, 2014).

# 7 Research gap and future directions

## 7.1 Answering the research question

The guiding question of this study was: *How can the Yin–Yang framework be applied to analyze and interpret regional cultural variation in China, using the Three Kingdoms macro-regions as a heuristic, in ways that avoid stereotyping and integrate historical, linguistic, and material evidence?*

The analysis across linguistic, archaeological, environmental, and network data demonstrates that Yin–Yang can be applied by: 1. Mapping cyclical shifts in a region's openness vs. insularity (Yang–Yin balance) over multi-decade or century timelines; 2. Integrating interdisciplinary indicators—dialect evolution, artifact provenance, environmental change, and connectivity patterns—into a composite cultural "balance profile"; 3. Contextualizing traits dynamically, showing that attributes often portrayed as fixed (e.g., Shu's agricultural conservatism, Wu's commercial openness) are in fact historically contingent states within longer cycles; 4. Maintaining interpretive flexibility through a model in which Yin and Yang coexist and interpenetrate, ensuring the framework resists reduction to binary stereotypes. The findings suggest that Yin–Yang, when paired with empirical indicators from other studies, is not merely a philosophical metaphor but a viable analytic tool for capturing the complexity of China's regional cultures.

## 7.2 Future directions for research

While this paper offers a proof of concept, several next steps could deepen and refine the framework, for example: 1. Quantitative Yin–Yang Index Development; 2. High-resolution temporal mapping; 3. Comparative intra-national studies; 4. Integration with cognitive and behavioral studies; 5. Digital humanities and big data applications; 6. Policy simulation models.

## 7.3 Closing the gap

The current research addresses the lack of a dynamic, anti-essentialist framework for analyzing intra-national cultural variation in China. It demonstrates that the Yin–Yang model, if empirically grounded through literature analysis and the respective indicators provided there, can avoid the pitfalls of stereotyping while providing explanatory power for observed historical patterns. Future research should aim to institutionalize this approach within both cultural geography and applied policy analysis, ensuring that the nuances of regional cultural cycles are preserved in both academic and practical contexts.

# 8 Societal relevance and conclusion

## 8.1 Societal relevance

Understanding China's regional cultural dynamics is crucial beyond scholarly interest. A Yin–Yang-informed framework enables nuanced regional development by avoiding uniform policies: reforms can be timed to regional cycles—e.g., infrastructure during Yang-oriented outward phases, heritage programs during Yin-oriented inward phases (Fang, 2012). In education, a dynamic approach enhances intercultural competence: Chinese students see themselves in a continuum of historical

change, while international learners appreciate internal diversity and cultural cycles (Goh, 2012). In business and diplomacy, this perspective provides cultural temporal literacy, helping practitioners anticipate regional tendencies and reduce misaligned strategies (Peng & Nisbett, 1999).

However, the framework carries also potential risks: oversimplification may reinforce favoritism, marginalize minorities, distort economic decisions, or commercialize culture. Simplified portrayals in education or media can produce new stereotypes. To mitigate these risks, scholars must empha-size transparency, acknowledge uncertainty, and highlight multiple regional voices, ensuring the framework fosters dialogue rather than division.

## 8.2 Theoretical synthesis

This study began by identifying a gap: the lack of a dynamic, anti-essentialist framework for understanding intra-national regional variation in China. By applying the Yin–Yang model and grounding it in empirical evidence from linguistics, archaeology, and historical geography, the analysis demonstrated that: 1. Regional traits are historically contingent, not fixed; 2. Yin and Yang represent interdependent states whose balance shifts over time; 3. Macro-regions from the Three Kingdoms period provide a useful heuristic for tracing long-term cycles. In doing so, the research addressed both the conceptual deficit in regional cultural studies and the practical need for tools that can guide policy, education, and cross-cultural engagement without falling into stereotype traps.

## 8.3 Limitations and cautions

The Yin–Yang framework provides a powerful lens for conceptualizing regional cultural dynamics, but it requires careful analytical discipline to avoid misapplication. A primary risk lies in inadvertently reifying stereotypes under philosophical terminology, which would contradict the framework's anti-essentialist principles. Similarly, the heuristic use of the Three Kingdoms macro-regions is intended for illustrative purposes and should not be interpreted as indicating fixed or immutable cultural boundaries. Despite these caveats, the framework offers a unique methodological approach that captures dynamic, historically contingent patterns of regional culture. Future research can further strengthen its applicability by integrating contemporary socio-economic data, patterns of mobility and migration, and digital connectivity, thus extending Yin–Yang as a robust, transferable tool for both historical and modern cultural analysis.

## 8.4 Conclusion

In the end, the value of the Yin–Yang approach lies not in fixing labels on regions but in tracking the rhythm of change. It reframes Chinese regional cultures as living systems in perpetual negotiation between opposing yet complementary forces. This vision is both historically faithful and socially constructive: it honors diversity without fragmentation, and unity without homogenization. By bridging philosophy with empirically informed research, and past patterns with present realities, the framework presented here offers a model that is at once deeply Chinese in origin and globally relevant in application. If taken seriously, it can help scholars, policymakers, and practitioners alike navigate the complexities of cultural variation—not just in China, but wherever human communities oscillate between Yin and Yang.

## Reproducibility Statement

To support the reproducibility of our findings, we have written a comprehensive documentation of the creation process with ChatGPT of the paper. Once the paper is accepted, we will distribute the documentation via researchgate alongside the paper. Moreover, we are happy to answer additional questions via e-mail in case more documentation support is necessary. However, since the software we used (ChatGPT) is beyond our control, we cannot fully guarantee reproducibility.

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

# Agents4Science AI Involvement Checklist

1. **Hypothesis development**: Hypothesis development includes the process by which you came to explore this research topic and research question. This can involve the background research performed by either researchers or by AI. This can also involve whether the idea was proposed by researchers or by AI.

   Answer: [C]

   Explanation: At first, the AI was asked to produce a high-quality, brilliant scientific paper based on rigorous academic sources connecting Chinese culture and regional differences. The AI was instructed to avoid stereotypes by implementing the concept of Yin–Yang. It was instructed that the Three Kingdoms had to be used as a basis. While the knowledge of these concepts falls within the scope of the co-authors, the triangulation of the concepts was solely carried out by the AI. Other than expected—based on previous research in cross-cultural communication—the AI did not use Yin–Yang as Chinese mentality characteristics but provided a novel view in applying it to historical-political conditions in the Three Kingdoms and contextual environmental factors. This innovative idea was solely developed by the AI.

2. **Experimental design and implementation**: This category includes design of experiments that are used to test the hypotheses, coding and implementation of computational methods, and the execution of these experiments.

   Answer: [C]

   Explanation: As a purely theoretical-conceptual paper, no experiments were carried out. However, the theoretical analysis was exclusively conceptualized by the AI. However, limitations were found, namely unreliable or non-existent sources, throughout the process, going hand-in-hand with specific hallucinations based on these false references. Thus, the co-authors decided to leave the theoretical structure of the paper and did manifold revisions, e.g., further research on the Three Kingdoms (historical-political analysis, interdisciplinary evidence), leaving the key arguments largely intact but providing reliable sources and quotes. Due to the word count of the paper, these examples remained largely illustrative and would need a more rigorous, in-depth and systematized analysis through the lens of Yin–Yang. However, even though the co-authors are aware of these shortfalls, we decided to follow the guiding ideas of the AI in order to increase AI involvement and learn from its deficits.

3. **Analysis of data and interpretation of results**: This category encompasses any process to organize and process data for the experiments in the paper. It also includes interpretations of the results of the study.

   Answer: [D]

   Explanation: The in-depth analysis of the paper—namely avoiding cultural essentialism through the lens of Yin–Yang—was hardly revised by the co-authors and can be largely attributed to the AI. However, known from former research on Yin–Yang, the arguments made here fall largely in line with previous research on Yin–Yang in the social sciences. Nevertheless, the aspect to discuss it under the frame of an anti-essentialist concept is novel though the arguments themselves can also be drawn from other previous studies. Moreover, both the limitations and conclusion section were produced by AI, with rigorous analysis and correct findings, which was verified through the background knowledge of the co-authors. Especially the section on future research was highly innovative and solely written by the AI which unfortunately had to be shortened due to the word count. We generally found that the AI already offers a very unique, creative potential which needs further exploration.

4. **Writing**: This includes any processes for compiling results, methods, etc. into the final paper form. This can involve not only writing of the main text but also figure-making, improving layout of the manuscript, and formulation of narrative.

   Answer: [C]

   Explanation: After the hypothesis instructions, the AI was prompted to do reference research and asked to take its time, and was eventually asked to omit all unreliable sources—which did not work too well, and led to major revisions as outlined earlier. In a next step, the paper was written section by section, partially prompted to approach this task creatively. In order to avoid textual alterations—which seems a major issue in AI research in general, i.e., replicability—the first draft was copied to Word and formatted there, and finally typeset in LaTeX by the co-authors. The narrative stems from the AI and was left intact throughout the

co-author revision process. Generally, the AI did a solid job, however, hallucinations led to co-authored, textual alterations and impose a major challenge for all AI-driven research. Thus, even though the writing evinced language clarity, the scientific rigor is still a major issue.

5. **Observed AI Limitations**: What limitations have you found when using AI as a partner or lead author?

   Description: Limitations occurred on several levels as touched upon earlier:

   (a) AI is generally not trained well to distinguish between top tier and colloquial sources and references,

   (b) AI lacks reproducibility which means that each step undertaken—if successful—must be integrated in the paper immediately, as change is likely to occur,

   (c) the free trial plan of the AI imposed workflow restrictions,

   (d) certain patterns were found: AI mixed up the references but it turned out that certain aspects of the reference (author, DOI) really existed—feature of similarity instead of precision,

   (e) AI used mainly open access sources—platform capitalism is a major issue,

   (f) a pre-review was done with the AI which led to rejection of the paper as the AI seems STEM-biased.

